# Research priorities for pregnancy hypertension: a UK priority setting partnership with the James Lind Alliance

Alison Ho [1], Louise Webster [1], Liza Bowen,[2] Fiona Creighton,[3] Sarah Findlay,[3] Chris Gale [4], Marcus Green [5], Toto Gronlund,[6] Laura A Magee [1], Richard J McManus [7], Hiten D Mistry [8], Gemma Singleton,[3] Jim Thornton [8], Rebecca Whybrow [1], Lucy Chappell [1]

For numbered affiliations see end of article.

**Correspondence to**
Dr Alison Ho;
alison.ho@kcl.ac.uk

## ABSTRACT

**Objectives** To identify research priorities for hypertensive disorders of pregnancy from individuals with lived experience and healthcare professionals.

**Design** Prospective surveys and consensus meetings using principles outlined by the James Lind Alliance.

**Setting** UK.

**Methods** A steering group was established and 'uncertainties' were gathered using an online survey and literature search. An interim online survey ranked long-listed questions and the top 10 research questions were reached by consensus at a final prioritisation workshop.

**Participants** Women, partners, relatives and friends of those with lived experience of pregnancy hypertension, researchers and healthcare professionals.

**Results** The initial online survey was answered by 278 participants (180 women with lived experience, 9 partners/relatives/friends, 71 healthcare professionals and 18 researchers). Together with a literature search, this identified 764 questions which were refined into 50 summary questions. All summary questions were presented in an interim prioritisation survey that was answered by 155 participants (87 women with lived experience, 4 partners/relatives/friends, 49 healthcare professionals and 15 researchers). The top 25 highest ranked questions were considered by the final prioritisation workshop. The top 10 uncertainties were identified by consensus and ranked as follows in order of priority: long-term consequences of pregnancy hypertension (for the woman and baby), short-term complications of pregnancy hypertension (for the woman and baby), screening tests for pre-eclampsia, prevention of long-term problems (for the woman and baby), causes of pregnancy hypertension, prevention of recurrent pregnancy hypertension, educational needs of healthcare professionals, diagnosis of pre-eclampsia, management of pregnancy hypertension, provision of support for women and families.

**Conclusions** Research priorities shared by those with lived experience of pregnancy hypertension and healthcare professionals have been identified. Researchers should use these to inform the choice of future studies in this area.

### Strengths and limitations of this study

► A consensus on research priorities in pregnancy hypertension was reached by those with lived experience and healthcare professionals.

► The approach used the James Lind Alliance methodology involving open access online surveys and a final face-to-face prioritisation meeting.

► Those with lived experience of pregnancy hypertension and healthcare professionals were involved at every stage of the priority setting partnership.

► The study may have been limited by an imbalance in ethnic diversity of those who participated despite efforts to optimise inclusion.

► The summary research questions are broad and may prove challenging for researchers to address within single studies.

## INTRODUCTION

Hypertensive disorders occur in up to 10% of all pregnancies[1] and include pre-eclampsia, gestational hypertension, chronic hypertension.[2] The pathophysiology differs to hypertension that occurs outside pregnancy and hypertensive disorders of pregnancy are all associated with adverse pregnancy outcomes,[3–7] but pre-eclampsia (hypertension and one or more of: proteinuria, acute kidney injury, liver dysfunction, neurological features, haemolysis, thrombocytopenia, fetal growth restriction[2]) has the most substantial impact on maternal and perinatal mortality and morbidity.[8] Half of women with pre-eclampsia deliver preterm and 1 in 20 stillbirths (without congenital abnormality) occur in women with pre-eclampsia.[9] Importantly, hypertensive disorders of pregnancy are also associated with an increased risk of long-term cardiovascular and metabolic morbidity and mortality for woman and child.[10–12]

Current research within hypertensive disorders of pregnancy is broad, exploring epidemiology, prediction, prevention, diagnosis, management and long-term implications for maternal and perinatal health. However, there is often a mismatch between research priorities identified by patients, clinicians and researchers.[13] [14] Areas for research prioritised by the American College of Obstetricians and Gynaecologists (ACOG),[15] International Society for the study of Hypertension in Pregnancy (ISSHP)[2] and the National Institute for Health and Care Excellence (NICE)[16] focus on different aspects, and the involvement of lay voices in these is often unclear.

The James Lind Alliance (JLA) facilitates priority setting partnerships (PSPs) so that an open dialogue among those with lived experience of a disorder, carers and clinician groups can occur in order to identify 'uncertainties' (questions which cannot be answered by existing research) that are important to all groups in a particular area of health.[17] Uncertainties are subsequently prioritised to ascertain the top 10 research questions, aiming to inform future research studies to address these questions. Since the establishment of the JLA in 2004, this methodology has been used to identify the top 10 research questions in areas such as asthma,[18] miscarriage[19] and hyperacusis.[20] Other JLAs have addressed research priorities in pregnancy complications such as preterm birth[21] and stillbirth[22] but these did not have a focus on hypertensive disorders of pregnancy. A Canadian PSP focused on hypertension, but pregnancy did not feature in their top 25 questions.[23] The JLA infrastructure is funded by the National Institute for Health Research.

## OBJECTIVE

To identify uncertainties and research priorities for hypertensive disorders of pregnancy in the UK from individuals with lived experience and healthcare professionals using the JLA methodology.[17]

## METHODS

The core steering group (LC, AH and LW) submitted a readiness questionnaire which was approved by the JLA Secretariat, based at the National Institute for Health Research Evaluation and the Trials and Studies Coordinating Centre, University of Southampton. A JLA advisor (TG) was assigned to facilitate the process and ensure that the JLA methodology was followed. We sought advice from our JLA advisor regarding ethical review prior to starting and concluded that, in line with other JLA PSPs, it was not required. Participants provided informed consent (indicated by completion of the survey and agreement to workshop attendance); it was made clear at each stage of the PSP that participation was voluntary, what participation involved, the purpose of the study and the use of data.

### The PSP stages

#### Initiation

Through peer knowledge and consultation, we formed a steering group for the PSP. Steering group meetings were chaired by TG (JLA advisor) and included lay members with lived experience of pregnancy hypertension and the CEO of a stake holding charity (GS, FC, SF and MG), obstetricians (JT, LC, LW and AH), an obstetric physician (LM), general practitioners (RJM and LB), a midwife (RW), a neonatologist (CG) and a research scientist (HM). The PSP lead was LC and information specialists were LW and AH. Women with lived experience and clinicians were represented at every stage and TG (as chair) was a neutral facilitator, ensuring a fair and transparent process with equal input from all groups. At the initial steering group meeting, the scope of the PSP was confirmed to include research priorities related to the following topics in the context of women with pregnancy hypertension: hypertensive disorders (including pre-eclampsia, gestational hypertension, chronic hypertension and white coat hypertension), women, babies, their partners and families, time period related to pregnancy (ie, pre-conception, antenatal, postnatal and long-term health outcomes), management related to pregnancy hypertension (ie, prevention, prediction, diagnosis and treatment), physical, social and emotional aspects, comorbidities such as renal disease or diabetes, genetics and information provision. The protocol was a published on the JLA website in July 2018 http://www.jla.nihr.ac.uk/priority-setting-partnerships/hypertension-in-pregnancy/downloads/Hypertension-in-Pregnancy-PSP-protocol.pdf

### Identifying clinical uncertainties

In October 2018, we launched an initial online survey to be answered by those with lived experience of hypertension in pregnancy and healthcare professionals (though we did not exclude the small number of responses submitted by researchers), using the Online Surveys platform.[24] Survey participants were asked to write up to three questions that they wanted answered by hypertension in pregnancy research. Additional optional questions included demographic details (gender, age range and ethnicity), name and preferred contact email. Contact details were collected only for the purposes of inviting participation in future activities related to the PSP and survey participants could remain anonymous. The survey was promoted through social media (Facebook, Twitter), clinical networks known to steering group members (targeting BAME (Black, Asian, and minority ethnic) and non-English speaking women) and the Action on Pre-eclampsia charity. In addition to potential uncertainties submitted through the online survey, the steering group identified uncertainties that had previously been reported by The ACOG, ISSHP and the NICE relevant to this topic.

### Refining uncertainties

All questions submitted (from the online survey and reported from ACOG, ISSHP and NICE) were assigned a unique question code. They were then reviewed by AH and LW and thematically grouped into nine categories: mechanisms, prediction, prenatal management, antenatal management, postnatal management, maternal

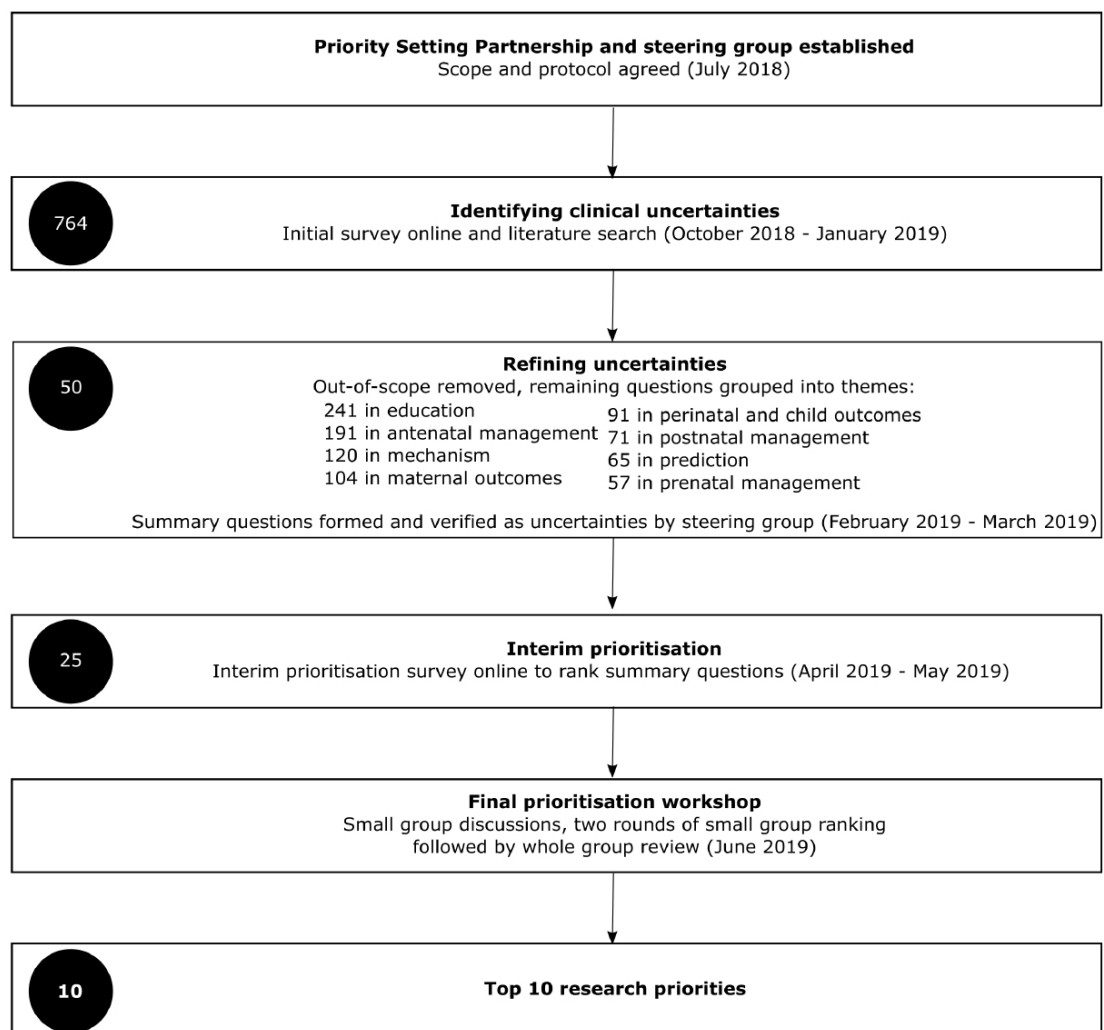

**Figure 1** Summary of the priority setting partnership stages. Number of questions at each stage illustrated in black circle.

outcomes, perinatal and child outcomes, education and out of scope. Submitted questions were assigned multiple themes, if applicable, and duplicate questions were removed. Each steering group member was assigned a theme and reviewed all questions within the theme to synthesise summary questions. A second steering group member reviewed the summary questions to ensure they were representative of the original questions and both members confirmed that the summary questions were not answered by existing research. All members of the steering group agreed the 50 summary questions to be put forward for interim prioritisation, based on being representative of the wider questions submitted, not answered by existing research, and ensuring that all themes were included.

### Interim prioritisation survey
A second survey was distributed in April 2019 using the same dissemination routes as the initial survey. The platform used was supplied by Optimal Workshop.[25] Survey participants were asked to identify the top 10 questions they felt to be most important from a randomly presented list of all summary questions. They were then asked to

identify their top three summary questions from within their top 10 so that further weighting could be applied to each question to identify the top 25 questions. Following closure of the survey in June 2019, the resulting highly ranked (based on frequency chosen) 25 questions were taken forward for final prioritisation. The source of each question was reviewed to ensure that questions from those with lived experience and clinicians were represented.

### Final prioritisation
The final prioritisation workshop took place in June 2019 and was chaired by three JLA advisors with oversight from some of the steering group. Participants had previously expressed their interest in taking part by submitting their contact details in either survey. Participants were representative of geographical diversity, and age, and included partners as well as those with lived experience.

Participants included 16 men and women with lived experience of pregnancy hypertension, 5 midwives, 4 obstetricians, 1 neonatologist, 1 general practitioner, a representative from the Stillbirth and Neonatal Death Charity (SANDs) and a representative from Best Beginnings charity. On the day, they were divided into three

groups, with equal numbers of participants with lived experience and clinicians in each, each chaired by a JLA advisor to ensure all participants were able to voice their opinions. In the first round of ranking, within each group, participants ranked the questions collectively after reflecting and discussing their reviews of priorities. Groups were subsequently reallocated with a different group composition for a second round of ranking, based on the combined ranking in the first round. A third and final priority setting session followed the aggregate ranking from the second round, which was presented to the whole group. The whole group discussed the results and reached a consensus on the final ranking with a focus on the top 10 prioritised uncertainties.

## Patient and public involvement

Patient and public involvement was a core part of the study from design, through all stages, to conclusion. From the outset, steering group lay members included those with lived experience of pregnancy hypertension (GS, FC, SF and MG). Both the initial survey and the interim prioritisation survey were answered by the public, the majority of whom had lived experience of pregnancy hypertension. Participants at the final prioritisation workshop included equal proportions of lay members (women with lived experience of pregnancy hypertension and their partners) and all others, including a representative from the SANDs and a representative from Best Beginnings charity.

## RESULTS

A summary of the PSP stages with a time line is shown in figure 1. The initial survey was answered by 278 participants, 65% of whom were women with lived experience of pregnancy hypertension and 26% of whom were healthcare professionals (table 1).

The initial survey and literature search conducted by the steering group identified 764 questions. Thematic review resulted in the greatest number of 241 questions in 'education', followed by 191 questions in 'antenatal management', 120 in 'mechanism', 104 in 'maternal outcomes', 91 in 'perinatal and child outcomes', 71 in 'postnatal management', 65 in 'prediction', 57 in 'prenatal management' and 16 out-of-scope questions. These out-of-scope questions were removed from further analysis. Review by the steering group resulted in the development of 50 summary questions. These were verified as uncertainties and all included in the interim prioritisation survey (listed in table 2).

The interim prioritisation survey was answered and completed by 155 people, 56% of whom were individuals with lived experience of pregnancy hypertension and 32% from healthcare professionals (table 3). The top 25 summary questions ranked at the final prioritisation workshop are listed in table 2. The results of the final top 10 prioritised and ranked uncertainties from the final prioritisation workshop are listed in table 4.

**Table 1** Characteristics of initial survey participants

| | Survey participants (n=278) N(%) |
|---|---|
| **Category selected** | |
| Women with lived experience of pregnancy hypertension | 180 (65) |
| Partner, relative or friend of someone with lived experience of pregnancy hypertension | 9 (3) |
| Healthcare professional | 71 (26) |
| Obstetrician | 22 (8) |
| Midwife | 27 (10) |
| General practitioner | 5 (2) |
| Paediatrician | 4 (1) |
| Neonatologist | 5 (2) |
| Physician | 2 (1) |
| Other | 6 (2) |
| Researcher | 18 (7) |
| **Demographic details** | |
| **Age** | |
| Less than 30 years | 27 (10) |
| 30–39 years | 105 (38) |
| 40–49 years | 75 (27) |
| 50–59 years | 54 (19) |
| 60 years and above | 15 (5) |
| No age selected | 2 (1) |
| **Gender** | |
| Female | 252 (91) |
| Male | 21 (8) |
| No gender selected | 5 (2) |
| **Ethnicity** | |
| White (British, Irish, other) | 239 (86) |
| Mixed | 5 (2) |
| Asian | 16 (6) |
| Chinese | 1 (<1) |
| Black | 12 (4) |
| Other ethnicity | 4 (1) |
| No ethnicity selected | 1 (<1) |

Values given as a number (percentage).

## DISCUSSION

### Statement of principal findings

In this PSP, we have identified the top 10 research priorities for hypertensive disorders of pregnancy incorporating the views of those with lived experience and healthcare professionals. Addressing these priorities will optimise understanding of short- and long-term complications of pregnancy hypertension for woman, their babies and wider families. It is noted that the top ten priorities

| Table 2 | Summary questions presented in second online survey for interim prioritisation (50 in total) |
|---|---|
| 1. | How can we optimise information giving for those at risk of or affected by pregnancy hypertension? |
| 2. | How can pregnancy hypertension (including pre-eclampsia) be prevented in a subsequent pregnancy? |
| 3. | What is the cause of pregnancy hypertension (including pre-eclampsia)? |
| 4. | What can be done prior to pregnancy to reduce the risk of pregnancy hypertension? |
| 5. | What is the best way to manage pre-eclampsia? |
| 6. | How can we provide better support for women with pregnancy hypertension and their families? |
| 7. | What is the best way to diagnose pre-eclampsia promptly? |
| 8. | What is the effectiveness and safety of antihypertensive agents at reducing blood pressure in women with pregnancy hypertension? |
| 9. | What are the long-term consequences of pre-eclampsia for the woman and baby? |
| 10. | What is the effectiveness and safety of pharmacological treatments once pre-eclampsia is diagnosed? |
| 11. | How does the placenta contribute to pre-eclampsia? |
| 12. | What is the optimal antihypertensive medication to use postnatally? |
| 13. | What is the best screening test for pre-eclampsia? |
| 14. | What are the optimum blood pressure thresholds (for initiation) and targets for antenatal antihypertensive treatment? |
| 15. | Is there a hereditary link in pre-eclampsia and are the risks different for daughters and sons after an affected pregnancy? |
| 16. | How can we predict complications of pregnancy hypertension (progression to pre-eclampsia)? |
| 17. | What is the optimal timing of delivery in women with pregnancy hypertension? |
| 18. | What interventions are effective and safe at reducing fetal growth restriction in women with pregnancy hypertension? |
| 19. | What are the long-term effects of pre-eclampsia on mental health? |
| 20. | Following pregnancy hypertension, what is the best way to prevent future long-term problems? |
| 21. | What are the educational needs of healthcare professionals managing women with pregnancy hypertension? |
| 22. | What are the fetal, infant and child outcomes in women taking antihypertensive agents? |
| 23. | How can we better prevent stillbirth in pre-eclampsia? |
| 24. | What are the optimum blood pressure thresholds (for initiation) and targets for postnatal antihypertensive treatment? |
| 25. | What prepregnancy management of women with chronic hypertension optimises pregnancy outcomes? |
| 26. | What is the optimal monitoring strategy for women before, during and after pregnancy hypertension (including in subsequent pregnancies)? |
| 27. | What are the risk factors for developing pregnancy hypertension and pre-eclampsia? |
| 28. | How can pregnancy hypertension (including pre-eclampsia) be prevented during a pregnancy? |
| 29. | What is the risk of pregnancy hypertension in a subsequent pregnancy? |
| 30. | What is the best test to predict pregnancy hypertension? |
| 31. | What methods are effective at measuring blood pressure in women with pregnancy hypertension (including self-monitoring, ambulatory, automated, manual)? |
| 32. | What is the paternal contribution to pre-eclampsia? |
| 33. | What are the characteristics of postpartum pre-eclampsia? |
| 34. | What is the safety of treatments for pregnancy hypertension for the fetus and infant? |
| 35. | What are the long-term effects of pre-eclampsia on cardiovascular disease for the woman and baby? |
| 36. | What are the long-term effects of pregnancy hypertension on subsequent maternal blood pressure? |
| 37. | What are the consequences of pregnancy hypertension on pre-eclampsia, birth weight and prematurity in that pregnancy? |
| 38. | What is the relationship between blood pressure in pregnancy and development of pregnancy hypertension? |
| 39. | What are the mechanisms for increased cardiovascular risk for a woman and her child? |
| 40. | What are the effects of lifestyle interventions (eg, diet, exercise) in reducing high blood pressure in pregnancy? |
| 41. | What are the short-term and long-term health implications for infants of women with pregnancy hypertension and can these be modified? |
| 42. | What are the long-term neurodevelopmental implications of pregnancy hypertension for the child? |
| 43. | How does pregnancy hypertension affect the growth of the baby? |
| 44. | What is the best way to follow up women who experience pregnancy hypertension? |
| 45. | Do pregnancy characteristics predict infant and child morbidity? |
| 46. | What is the effectiveness and safety of aspirin for prevention of pre-eclampsia? |

Continued

**Table 2  Continued**

| 47. | What are the links between maternal emotional well-being and pregnancy hypertension? |
| 48. | What non-pharmacological treatments are effective in treating high blood pressure following pregnancy hypertension? |
| 49. | How do sleep disorders affect pregnancy hypertension? |
| 50. | What are patient-reported outcomes of interest related to hypertension in pregnancy? |

The first 25 listed above were highly ranked in the survey and therefore brought forward to the final prioritisation workshop.

**Table 3  Characteristics of interim prioritisation survey participants**

| | Survey participants (n=155) N(%) |
| --- | --- |
| **Category** | |
| Women with lived experience of pregnancy hypertension | 87 (56) |
| Partner, relative or friend of someone with lived experience of pregnancy hypertension | 4 (3) |
| Healthcare professional | 49 (32) |
| Obstetrician | 21 (14) |
| Midwife | 14 (9) |
| General practitioner | 3 (2) |
| Paediatrician or neonatologist | 2 (1) |
| Physician | 4 (3) |
| Neonatal nurse | 2 (1) |
| Other | 3 (2) |
| Researcher | 15 (10) |
| **Demographic details** | |
| **Age** | |
| Less than 30 years | 10 (6) |
| 30–39 years | 63 (41) |
| 40–49 years | 45 (29) |
| 50–59 years | 21 (14) |
| 60 years and above | 9 (6) |
| No age selected | 7 (5) |
| **Gender** | |
| Female | 133 (86) |
| Male | 14 (9) |
| No gender selected | 8 (5) |
| **Ethnicity** | |
| White (British, Irish, other) | 130 (84) |
| Mixed | 3 (2) |
| Asian | 7 (5) |
| Chinese | 1 (<1) |
| Black | 4 (3) |
| No ethnicity selected | 10 (7) |

Values given as a number (percentage).

encompass the range of outstanding challenges in this field, including improving screening, prevention and management, addressing both short-term and long-term complications, and the mental health consequences (as well as the physical health consequences). Summary questions relating to education and information giving, and provision of support, were highly prioritised throughout the process and their presence in the top 10 research priorities reflects this. These research priorities provide a clear steer to funding bodies for the future awards.

### Strengths and weaknesses of the study

To our knowledge, this is the first national PSP for hypertensive disorders of pregnancy to inform the direction of future research in this area. We have adhered to the JLA methodology, including prospective publication of our protocol http://www.jla.nihr.ac.uk/priority-setting-partnerships/hypertension-in-pregnancy/. Discussions at the final prioritisation workshop were facilitated by experienced JLA advisors to ensure that no group or individual dominated the decision making. However, it is possible that participants may have been biased in their prioritisation, for example knowledge of existing research projects that may answer certain questions (and therefore giving a lower rating), or that further research was still needed (and therefore giving a higher priority). The PSP has illustrated a need for a multidisciplinary and holistic approach when caring for women with pregnancy hypertension. Women and partners with lived experience of pregnancy hypertension were included from the outset and at every stage (GS, FC, SF and MG), so that our approach to the PSP optimised their participation. The large proportion of survey responses from those with lived experience and participation in the final workshop reflects this.

The initial survey and interim prioritisation survey were only available online; this may have been a barrier to participation, but women with recent lived experience of pregnancy hypertension (of reproductive age) have high rates of access to such survey methods. Despite efforts to reach an ethnically diverse population for survey responses, the number of participants from Black and Asian minority ethnic groups was low. The priorities are broad and thus translation into high-quality quantitative and qualitative studies to answer them may require further work.

| Table 4 | Final top 10 prioritised and ranked uncertainties |
|---|---|
| **Priority** | **Research question** |
| 1. | What are the long-term physical and mental health consequences of pregnancy hypertension (including pre-eclampsia) for the woman, baby and family? |
| 2. | How can we predict and prevent shorter term complications of pregnancy hypertension (including stillbirth, fetal growth restriction, neonatal death, progression to pre-eclampsia)? |
| 3. | What is the best screening test for pre-eclampsia? |
| 4. | Following pregnancy hypertension, what is the best way to prevent future long-term problems? |
| 5. | What is the cause of pregnancy hypertension (including pre-eclampsia)? |
| 6. | How can pregnancy hypertension (including pre-eclampsia) be prevented in a subsequent pregnancy? |
| 7. | What are the educational needs of healthcare professionals managing women with pregnancy hypertension? |
| 8. | What is the best way to diagnose pre-eclampsia promptly? |
| 9. | What is the best way to manage pregnancy hypertension (including optimal antenatal and postnatal antihypertensive medication and optimal timing of delivery)? |
| 10. | How can we provide better support for women with pregnancy hypertension and their families? |

## Strengths and weaknesses in relation to other studies

The final top 10 prioritised and ranked uncertainties encompass all uncertainties reported by ACOG, ISSHP and NICE and thus reflect the overlapping uncertainties important to those with lived experience of pregnancy hypertension and healthcare professionals. The preterm birth [26]PSP had an overlapping uncertainty of, 'Which treatments are most effective to prevent early onset pre-eclampsia.' Despite the lifelong impact of pregnancy hypertension on cardiovascular disease, there were no pregnancy hypertension questions in the final top 10 hypertension Canada PSP[23] and thus this PSP reflects a set of research priorities specific to pregnancy hypertension. As seen with other PSPs,[19 27 28] the need for improved education and support has been highlighted for further research, strongly endorsed by the lay participants. All of the final questions posed were derived from both lay and healthcare professionals as the JLA chair ensured even contribution throughout. No substantial mismatch in questions posed by those with lived experience and clinicians/researchers was identified in this PSP.

Our final prioritisation workshop required participants to attend in person and this may have been a barrier to some of those with lived experience of pregnancy hypertension. We minimised attrition due to the requirement for childcare by welcoming babies in arms and making childcare reimbursable. Further inclusion through video conferencing may have enabled more participants to attend including those as hospital inpatients[29]; however, remote working may have impacted on the dynamics of the final workshop.

## Meaning of the study with possible implications for clinicians and policy-makers

The list of research priorities provides guidance for researchers for future study topic choice within hypertensive disorders of pregnancy and should inform funding body decisions. While most of the identified areas for research overlap with current broad research themes, the study has highlighted a specific need to optimise public information giving and education for hypertensive disorders of pregnancy that might not otherwise have been so clearly recognised as a priority particular from those with lived experience.

## Unanswered questions and future research

All uncertainties listed remain unanswered by existing research, reflecting gaps in our knowledge of pregnancy hypertension. Further work to refine each research priority into formatted research questions (eg, using the Population, Intervention, Comparator, Outcome framework) would enable researchers to answer them effectively. We anticipate that our findings will encourage researchers to address these priorities important to both those with lived experience of pregnancy hypertension and healthcare professionals.

**Author affiliations**
[1]Women and Children's Health, School of Life Course Sciences, King's College London, London, UK
[2]School of Population Health and Environmental Sciences, King's College London, London, UK
[3]Lay member, London, UK
[4]Neonatal Medicine, School of Public Health, Imperial College London, London, UK
[5]Action on Pre-eclampsia, The Stables, Evesham, UK
[6]James Lind Alliance, National Institute for Health Research Evaluation, Trials and Studies Coordinating Centre, Southampton, UK
[7]Dept of Primary Care Health Sciences, University of Oxford, Oxford, UK
[8]Division of Child Health, Obstetrics and Gynaecology, School of Medicine, University of Nottingham, Nottingham, UK

**Acknowledgements** We thank all those who participated in the initial survey, interim prioritisation survey and final prioritisation workshop.

**Contributors** LC, AH and LW made the application to the James Lind Alliance for a pregnancy hypertension priority setting partnership. AH and LW reviewed and coded all submissions from the initial survey. AH, LW, GS, MG, FC, SF, RW, JT, CG, RJM, LB, HM, LM, TG and LC contributed to the protocol design, production of both the initial survey and interim prioritisation survey, promotion and dissemination of surveys to

partner organisations and formation of summary questions. AH, LW and LC drafted the manuscript. All authors reviewed and approved the final manuscript before submission.

**Funding** This work was funded by the national Institute for Health Research (NIHR) Research Professorship (Chappell; RP-2014-05-019). Funds supporting this project were used for JLA fees and to support the running costs of the project (including childcare costs for PPIE members, travel expenses). No salaries were provided to the research team for this project.

**Disclaimer** The views expressed are those of the authors and not necessarily those of the UK National Health Service, the National Institute for Health Research, or the Department of Health and Social Care.

**Competing interests** LC reports grants from the National Institute for Health Research during the conduct of the study. CG reports grants from Medical Research Council during the conduct of the study; grants from National Institute for Health Research, Mason Medical Research foundation, Canadian Institute for Health Research, Rosetrees Foundation, grants and personal fees from Chiesi Pharmaceuticals, outside the submitted work. HM reports grants from the British Heart Foundation. RJM reports grants from NIHR, grants from Stroke Association, outside the submitted work; and has received BP monitors for research from Omron. He occasionally receives travel expenses/honoraria for speaking at conferences. The latter are paid to Green Templeton College Oxford. All additional interests are outside the direct remit of the submitted work. All other authors declare no competing interests.

**Patient and public involvement** Patients and/or the public were involved in the design, or conduct, or reporting, or dissemination plans of this research. Refer to the Methods section for further details.

**Patient consent for publication** Not required.

**Provenance and peer review** Not commissioned; externally peer reviewed.

**Data availability statement** Data are available in a public, open access repository. Data regarding the source of all summary questions (including the top 10 research priorities) are available from http://www.jla.nihr.ac.uk/priority-setting-partnerships/hypertension-in-pregnancy/

**ORCID iDs**
Alison Ho http://orcid.org/0000-0003-3293-4476
Louise Webster http://orcid.org/0000-0001-9050-4242
Chris Gale http://orcid.org/0000-0003-0707-876X
Marcus Green http://orcid.org/0000-0002-4561-8256
Laura A Magee http://orcid.org/0000-0002-1355-610X
Richard J McManus http://orcid.org/0000-0003-3638-028X
Hiten D Mistry http://orcid.org/0000-0003-2564-7348
Jim Thornton http://orcid.org/0000-0001-9764-6876
Rebecca Whybrow http://orcid.org/0000-0001-7484-1492
Lucy Chappell http://orcid.org/0000-0001-6219-3379

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
