## [Reviewer comments · BMJ Open]

ARTICLE DETAILS

TITLE (PROVISIONAL)	Research Priorities for Pregnancy Hypertension: a UK Priority Setting Partnership with the James Lind Alliance
AUTHORS	Ho, Alison; Webster, Louise; Bowen, Liza; Creighton, Fiona; Findlay, Sarah; Gale, Chris; Green, Marcus; Gronlund, Toto; Magee, Laura; McManus, Richard; Mistry, Hiten; Singleton, Gemma; Thornton, Jim; Whybrow, rebecca; Chappell, Dr Lucy

VERSION 1 – REVIEW

REVIEWER	Natalie A Bello Columbia University
REVIEW RETURNED	24-Dec-2019

GENERAL COMMENTS	I applaud the authors for this novel priority setting partnership that was inclusive of many stakeholders and identified research priorities for hypertensive disorders of pregnancy. The major limitation, as they point out, is a lack of diversity in their sample. I also found the manuscript to be a bit long and redundant at times. With minor revisions it will be an important contribution to the literature.
--

REVIEWER	Nancy Nixon Tom Baker Cancer Center Calgary, AB Canada
REVIEW RETURNED	13-Feb-2020

GENERAL COMMENTS	It is not clear what 'from a lay and clinical perspective' means in the objective statement. Could this be changed to reflect 'from the patients/affected individuals, and healthcare professionals?' I think it would be worthwhile to focus a bit less on the methods, and address more in the discussion what we saw in the 'top 10'. They did mention that education was highlighted, but more discussion of the top 10 uncertainties would be interesting.
---

REVIEWER	Rhys Thomas Newcastle University, UK
REVIEW RETURNED	02-Apr-2020

GENERAL COMMENTS	This paper is one of many that use the robust methodology of the James Lind Alliance and is therefore a model of good practice for participation research with experts with lived experience and their clinicians. Two more important areas for thought
--

	Can you give speciality specific examples as to why this was needed for hypertension in pregnancy and how this top 10 will change this area in particular? Do you want to consider the role of implicit / explicit biases being introduced at the face to face meeting? How was consensus achieved? Additional areas for thought The method for each PSP can differ Here they used researchers (n=18) for question generation, rather than just clinicians and patients It would be really neat if they could identify a mismatch in lay/clinician research priorities from within their speciality (in addition to references 12, 13) Do you want to comment on the optimal make-up of your steering group? JLA chair rather than a joint lay chair, joint clinical chair? 4 lay members and 6 clinicians, one researcher? Were there charity or patient groups who could have been invited too? How were lay members identified? Had any suffered fetal loss? The hyperlinked pdf on page 8 could be a reference You don't need http:// in a reference (really don't need the www. either) Why the limit of three questions? Table 2 could be a supplementary table What were your efforts to reach BAME participants? Was the funding to the research team or to the JLA for this?
--	--

VERSION 1 – AUTHOR RESPONSE

Reviewer #1:

1.1 I applaud the authors for this novel priority setting partnership that was inclusive of many stakeholders and identified research priorities for hypertensive disorders of pregnancy. The major limitation, as they point out, is a lack of diversity in their sample. I also found the manuscript to be a bit long and redundant at times. With minor revisions it will be an important contribution to the literature.
A. We have reviewed the manuscript as a whole and edited to ensure that we have sufficiently covered the methodology but with the aim of keeping it succinct. We have aimed to avoid redundant text wherever possible.

Reviewer #2:

2.1 It is not clear what 'from a lay and clinical perspective' means in the objective statement. Could this be changed to reflect 'from the patients/affected individuals, and healthcare professionals?'
A. We have changed the objectives in lines 27-28 and 97-99:
'To identify uncertainties and research priorities for hypertensive disorders of pregnancy in the United Kingdom from individuals with lived experience and healthcare professionals using the JLA methodology.'

2.2 I think it would be worthwhile to focus a bit less on the methods, and address more in the discussion what we saw in the 'top 10'. They did mention that education was highlighted, but more discussion of the top 10 uncertainties would be interesting.

A. We have altered the Statement of principal findings and Meaning of the study sections in the Discussion accordingly:

Lines 215-222:

'In this priority setting partnership we have identified the top ten research priorities for hypertensive disorders of pregnancy incorporating the views of those with lived experience and healthcare professionals. Addressing these priorities will optimise understanding of short- and long-term complications of pregnancy hypertension for woman, their babies and wider families. It is noted that the top ten priorities encompass the range of outstanding challenges in this field, including improving screening, prevention and management, addressing both short and long-term complications, and the mental health consequences (as well as the physical health consequences). Summary questions relating to education and information giving, and provision of support, were highly prioritised throughout the process and their presence in the top ten research priorities reflects this.'

Lines 267-272:

'The list of research priorities provides guidance for researchers for future study topic choice within hypertensive disorders of pregnancy and should inform funding body decisions. Whilst most of the identified areas for research overlap with current broad research themes, the study has highlighted a specific need to optimise public information giving and education for hypertensive disorders of pregnancy that might not otherwise have been so clearly recognised as a priority particular from those with lived experience.'

Reviewer #3:

This paper is one of many that use the robust methodology of the James Lind Alliance and is therefore a model of good practice for participation research with experts with lived experience and their clinicians.

Two more important areas for thought:

3.1 Can you give speciality specific examples as to why this was needed for hypertension in pregnancy and how this top 10 will change this area in particular?

A. Introduction altered lines 72-97:

'Hypertensive disorders occur in up to 10% of all pregnancies 1 and include pre-eclampsia, gestational hypertension, chronic hypertension. 2 The pathophysiology differs to hypertension that occurs outside pregnancy and hypertensive disorders of pregnancy are all associated with adverse pregnancy outcomes, 3-7 but pre-eclampsia (hypertension and one or more of: proteinuria, acute kidney injury, liver dysfunction, neurological features, haemolysis, thrombocytopenia, fetal growth restriction 2) has the most substantial impact on maternal and perinatal mortality and morbidity. 8 Half of women with pre-eclampsia deliver preterm and one in twenty stillbirths (without congenital abnormality) occur in women with pre-eclampsia. 9 Importantly, hypertensive disorders of pregnancy are also associated with an increased risk of long-term cardiovascular and metabolic morbidity and mortality for woman and child. 10,11

Current research within hypertensive disorders of pregnancy is broad, exploring epidemiology, prediction, prevention, diagnosis, management and long-term implications for maternal and perinatal health. However, there is often a mismatch between research priorities identified by patients, clinicians and researchers. 12,13 The James Lind Alliance (JLA) facilitates priority setting partnerships (PSPs) so that an open dialogue amongst those with lived experience of a disorder, carers and clinician groups can occur in order to identify "uncertainties" (questions which cannot be answered by existing research) that are important to all groups in a particular area of health. 14 Uncertainties are subsequently prioritised to ascertain the top 10 research questions, aiming to inform future research studies to address these questions. Since the establishment of the JLA in 2004, this methodology has been used to identify the top 10 research questions in areas such as asthma 15, miscarriage 16 and hyperacusis. 17 Other JLAs have addressed research priorities in pregnancy

complications such as preterm birth and stillbirth but these did not have a focus on hypertensive disorders of pregnancy. A Canadian priority setting partnership focussed on hypertension, but pregnancy did not feature in their top 25 questions. The JLA infrastructure is funded by the National Institute for Health Research (NIHR).'

Discussion altered lines 215-224:

'In this priority setting partnership we have identified the top ten research priorities for hypertensive disorders of pregnancy incorporating the views of those with lived experience and healthcare professionals. Addressing these priorities will optimise understanding of short- and long-term complications of pregnancy hypertension for woman, their babies and wider families. It is noted that the top ten priorities encompass the range of outstanding challenges in this field, including improving screening, prevention and management, addressing both short and long-term complications, and the mental health consequences (as well as the physical health consequences). Summary questions relating to education and information giving, and provision of support, were highly prioritised throughout the process and their presence in the top ten research priorities reflects this. These research priorities provide a clear steer to funding bodies for the future awards.'

3.2 Do you want to consider the role of implicit / explicit biases being introduced at the face to face meeting? How was consensus achieved?

A. Methods altered lines 175-177:

'On the day they were divided into three groups, with equal numbers of participants with lived experience and clinicians in each, each chaired by a JLA advisor to ensure all participants were able to voice their opinions.'

Discussion lists biases of the face to face meeting as a limitation in lines 229-234:

'Discussions at the final prioritisation workshop were facilitated by experienced JLA advisors to ensure that no group or individual dominated the decision making. However, it is possible that participants may prioritise based on different criteria, such as either considering that existing research may answer the question (and therefore giving a lower rating), or that further research was still needed (and therefore giving a higher priority).'

3.3 The method for each PSP can differ. Here they used researchers (n=18) for question generation, rather than just clinicians and patients.

A. Methods clarified lines 131-133:

'In October 2018, we launched an initial online survey to be answered by those with lived experience of hypertension in pregnancy and healthcare professionals (though we did not exclude the small number of responses submitted by researchers), using the <http://www.onlinesurveys.ac.uk> platform. Survey participants were asked to write up to three questions that they wanted answered by hypertension in pregnancy research.'

3.4 It would be really neat if they could identify a mismatch in lay/clinician research priorities from within their speciality (in addition for references 12, 13)

A. Discussion altered lines 256-259:

'All of the final questions posed were derived from both lay and healthcare professionals as the JLA chair ensured even contribution throughout. No substantial mismatch in questions posed by those with lived experience and clinicians/researchers was identified in this priority setting partnership.'

3.5 Do you want to comment on the optimal make-up of your steering group? JLA chair rather than a joint lay chair, joint clinical chair? 4 lay members and 6 clinicians, one researcher? Were there charity or patient groups who could have been invited too? How were lay members identified? Had any suffered fetal loss?

A. Methods altered lines 114-118:

'Steering group meetings were chaired by TG (JLA advisor) and included lay members with lived experience of pregnancy hypertension and the CEO of a stake holding charity (GS, FC, SF, MG), obstetricians (JT, LC, LW and AH), an obstetric physician (LM), general practitioners (RM, LB), a midwife (RW), a neonatologist (CG), and a research scientist (HM). The Priority Setting Partnership lead was LC and information specialists were LW and AH.'

3.6 The hyperlinked pdf on page 8 could be a reference. You don't need http:// in a reference (really don't need the www. either).

A. Methods altered line 160

'The platform used was supplied by Optimal Workshop.'

3.7 Why the limit of three questions?

A. Methods expanded lines 162-164

'They were then asked to identify their top three summary questions from within their top 10 so that further weighting could be applied to each question to identify the top 25 questions.'

3.8 Table 2 could be a supplementary table

A. We are happy to take editorial advice on this. We think that it is useful to have the longer list of top 50 questions to demonstrate the breadth of topics but can present as the editor wishes.

3.9 What were your efforts to reach BAME participants?

A. Methods altered line 138-140:

'The survey was promoted through social media (Facebook, Twitter), clinical networks known to steering group members (targeting BAME and non-English speaking women) and the Action on Pre-eclampsia charity (APEC).'

3.10 Was the funding to the research team or to the JLA for this?

A. Funding statement edited to read:

Funds supporting this project were used for JLA fees and to support the running costs of the project (including childcare costs for PPIE members, travel expenses). No salaries were provided to the research team for this project.

VERSION 2 – REVIEW

REVIEWER	Rhys Thomas Newcastle University
REVIEW RETURNED	10-May-2020

GENERAL COMMENTS	The authors have done a commendable job of addressing the questions posed by all three reviewers. A couple of my points (reviewer 3) probably could have been clearer. In answering 3.4 (about clinician / lay conflict) - I was in fact speaking to the need for the study in the first place - are there areas of conflict from within their field that demonstrate the need for a PSP? Secondly I was more interested in why they chose to set up their PSP in that format rather than which individual was the chair etc. But these are not serious impediments to publishing this paper which will be a fine addition to the literature.
---

VERSION 2 – AUTHOR RESPONSE

Response to the Editor and Reviewers' Comments

1. Please move the Patient and Public Involvement statement to the end of the methods section

Done

2. The meta-data for this manuscript states that participants provided informed consent to participate in this study. Please add this information to your manuscript.

We have added the following:

Methods (line 113)

Participants provided informed consent (indicated by completion of the survey and agreement to workshop attendance); it was made clear at each stage of the priority setting partnership that participation was voluntary, what participation involved, the purpose of the study and the use of data.

Reviewer: 3

The authors have done a commendable job of addressing the questions posed by all three reviewers. A couple of my points (reviewer 3) probably could have been clearer.

1. In answering 3.4 (about clinician / lay conflict) - I was in fact speaking to the need for the study in the first place - are there areas of conflict from within their field that demonstrate the need for a PSP?

A. We have clarified this by adding the following:

Introduction (line 86)

Areas for research prioritised by The American College of Obstetricians and Gynaecologists (ACOG), International Society for the study of Hypertension in Pregnancy (ISSHP) and the National Institute for Health and Care Excellence (NICE) focus on different aspects, and the involvement of lay voices in these is often unclear.

2. Secondly I was more interested in why they chose to set up their PSP in that format rather than which individual was the chair etc.

A. We applied the JLA methodology and have addressed this as follows:

Introduction (line 90)

The James Lind Alliance (JLA) facilitates priority setting partnerships (PSPs) so that an open dialogue amongst those with lived experience of a disorder, carers and clinician groups can occur in order to identify “uncertainties” (questions which cannot be answered by existing research) that are important to all groups in a particular area of health.¹⁵

Methods (line 124)

Women with lived experience and clinicians were represented at every stage and TG (as chair) was a neutral facilitator, ensuring a fair and transparent process with equal input from all groups